# Long-distance electron transport in multicellular freshwater cable bacteria

Tingting Yang[1†], Marko S Chavez[1†], Christina M Niman[1], Shuai Xu[1], Mohamed Y El-Naggar[1,2,3]*

[1]Department of Physics and Astronomy, University of Southern California, Los Angeles, United States; [2]Molecular and Computational Biology Section, Department of Biological Sciences, University of Southern California, Los Angeles, United States; [3]Department of Chemistry, University of Southern California, Los Angeles, United States

*For correspondence:
mnaggar@usc.edu

[†]These authors contributed equally to this work

Competing interest: The authors declare that no competing interests exist.

**Abstract** Filamentous multicellular cable bacteria perform centimeter-scale electron transport in a process that couples oxidation of an electron donor (sulfide) in deeper sediment to the reduction of an electron acceptor (oxygen or nitrate) near the surface. While this electric metabolism is prevalent in both marine and freshwater sediments, detailed electronic measurements of the conductivity previously focused on the marine cable bacteria (*Candidatus* Electrothrix), rather than freshwater cable bacteria, which form a separate genus (*Candidatus* Electronema) and contribute essential geochemical roles in freshwater sediments. Here, we characterize the electron transport characteristics of *Ca*. Electronema cable bacteria from Southern California freshwater sediments. Current–voltage measurements of intact cable filaments bridging interdigitated electrodes confirmed their persistent conductivity under a controlled atmosphere and the variable sensitivity of this conduction to air exposure. Electrostatic and conductive atomic force microscopies mapped out the characteristics of the cell envelope's nanofiber network, implicating it as the conductive pathway in a manner consistent with previous findings in marine cable bacteria. Four-probe measurements of microelectrodes addressing intact cables demonstrated nanoampere currents up to 200 μm lengths at modest driving voltages, allowing us to quantify the nanofiber conductivity at 0.1 S/cm for freshwater cable bacteria filaments under our measurement conditions. Such a high conductivity can support the remarkable sulfide-to-oxygen electrical currents mediated by cable bacteria in sediments. These measurements expand the knowledgebase of long-distance electron transport to the freshwater niche while shedding light on the underlying conductive network of cable bacteria.

## eLife assessment

This work presents **fundamental** new insights into the conductivity of freshwater cable bacteria. The evidence supporting the conclusions, which was collected using appropriate techniques, is **compelling**. The work will be of interest to environmental microbiologists and the microbial electrochemistry community.

## Introduction

Electron transfer reactions are fundamental to biological energy conversion. Respiratory organisms, for instance, gain energy by routing electron flow from an electron donor to a terminal electron acceptor through the network of reduction-oxidation (redox) cofactors that constitute the cellular electron transport chain (*Gray and Winkler, 2003*). While the length scales of biological electron transfer events were long thought to be limited to nanometer distances and confined within individual

cells, recent observations of fast long-distance extracellular and intercellular electron transport in microbial communities have upended this view (*Atkinson et al., 2023*). In fact, with the discovery of micron-scale electron transport via cytochrome nanowires from metal-reducing bacteria (*Wang et al., 2019*; *Wang et al., 2022*), conductive bacterial biofilms (*Yates et al., 2016*; *Xu et al., 2018*), inter-species electron transfer in microbial consortia (*McGlynn et al., 2015*), and multicellular cable bacteria (CB) (*Pfeffer et al., 2012*), the length scales of microbial electron transport observations have jumped from the nanometer to centimeter scale during the last 15 years.

The novel multicellular filamentous CB, which belong to the *Desulfobulbaceae* family of Deltaproteobacteria (*Pfeffer et al., 2012*), are a particularly interesting system for understanding the limits of biological electron transport given that they are the only known living structures capable of macroscopic (up to centimeter-scale) electron conduction. This macroscopic electron transport allows CB to gain energy from coupling sulfide oxidation by cells that dwell in the deeper sulfidic zone to oxygen reduction by cells near the surface sediment in both marine and freshwater sediments (*Pfeffer et al., 2012*; *Nielsen et al., 2010*; *Meysman et al., 2015*; *Risgaard-Petersen et al., 2015*). The energy harnessed from this process allows CB to form a dense matrix (up to hundreds of meters per cm$^2$) that composes a majority of the biomass in sediment when blooming (*Marzocchi et al., 2018*; *Schauer et al., 2014*; *Malkin et al., 2014*; *Burdorf et al., 2016*; *van de Velde et al., 2016*). While the molecular pathway of the conductive network responsible for centimeter-scale electron transport along the cables is not yet understood, the genomic basis and components allowing CB to carry out the two spatially-separated half-reactions are just beginning to come into focus. CB appear to oxidize sulfide by reversing the canonical sulfate reduction pathway and are capable of sulfur disproportionation, while oxygen reduction is hypothesized to be driven by periplasmic cytochromes without energy conservation due to the absence of terminal oxidases (*Müller et al., 2020*; *Kjeldsen et al., 2019*). The electrogenic sulfur oxidation and its coupling to distant redox events by CB significantly affect sulfur cycling and other essential geochemical processes, including nitrogen and metal cycling in global sediments (*Kjeldsen et al., 2019*; *Marzocchi et al., 2014*).

But what is the conductive pathway responsible for long-distance electron transport along an entire CB filament? The answer appears to lie in the unusual cell envelope of CB, where up to thousands of cells share a common outer membrane and parallel longitudinal ridges run along the surface of the entire filament (*Pfeffer et al., 2012*; *Cornelissen et al., 2018*). These ridges reflect the presence of a network of long nanofibers (33–67 nm diameter) along the entire length of each cable in the periplasmic space between the common outer membrane and the cellular inner membranes (*Cornelissen et al., 2018*; *Leonid et al., 2023*). Initial electrostatic force microscopy (EFM) of these ridges showed striking electrostatic contrast (*Pfeffer et al., 2012*), implicating the periplasmic nanofibers as the current-carrying structures in marine CB filaments. More recently, the conductivity of intact air-dried marine CB filament and chemically extracted sheaths containing the nanofiber network was measured directly by various electronic techniques (*Meysman et al., 2019*; *Thiruvallur Eachambadi et al., 2020*; *Bonné et al., 2020*). Tens of nanoampere currents could be observed along both intact cable filaments and extracted nanofiber sheaths spanning hundreds of microns between microelectrodes in N$_2$/vacuum environments, but the conductance declined rapidly upon exposure to ambient air (*Meysman et al., 2019*). Conductive atomic force microscopy (C-AFM) with stiff tips that scrape outer cell surface layers revealed that the conductive path indeed correlates with the network of individual periplasmic nanofibers (*Thiruvallur Eachambadi et al., 2020*). For the marine CB nanofibers, conductivities spanning the $10^{-2}$–$10^1$ S/cm range, with the upper end reaching as high as 79 S/cm, have been estimated (*Meysman et al., 2019*) these values represent a record high for any biological structure. Furthermore, temperature-dependent electronic measurements suggest a thermally activated Arrhenius-type charge transport process with a low activation energy, suggesting that the CB conductive properties are comparable to organic semiconductor materials (*Bonné et al., 2020*). The composition of the CB conductive nanofibers remains unknown, but recent evidence points to yet-unknown core proteins rich in sulfur-ligated nickel cofactors that mediate electron transport through an enigmatic physical mechanism, and that the conductivity is diminished upon oxidation of the Ni/S groups (*Boschker et al., 2021*).

On the basis of 16S rRNA gene sequencing, the CB clade consists of two novel candidate genera: the mostly marine *Candidatus* Electrothrix and mostly freshwater *Candidatus* Electronema (*Trojan et al., 2016*). In addition, a more distant group of CB has been identified in groundwater environments

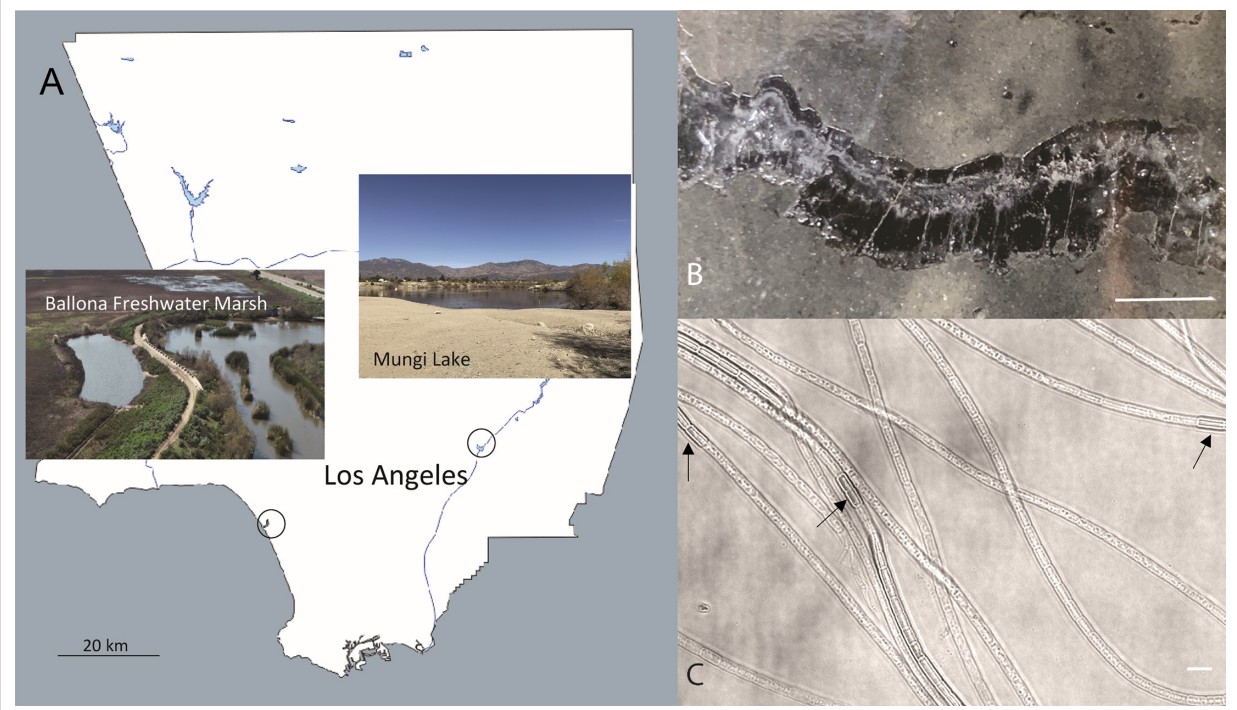

**Figure 1.** The sampling locations in Southern California and optical microscopy images of intact freshwater cable bacteria. (**A**) Sampling sites for cable bacteria from the Ballona Freshwater Marsh and Mungi Lake, CA, USA. (**B**) Enrichment in laboratory incubations resulted in a high density of long cable bacteria filaments, some of which could be observed directly in cracked sediments. Scale bar: 35 mm. (**C**) Optical microscopy revealed the characteristic end-to-end multicellular morphology of the cable bacteria filaments, revealing occasional darker cells (black arrows) with slightly larger diameters, consistent with 'empty cage' cells scaffolded by the nanofiber network of cable bacteria (see *Figure 2*). Scale bar: 5 μm.

(*Müller et al., 2020*; *Müller et al., 2016*). However, the abovementioned electronic characterization of CB conduction largely focused on marine CB, with the exception of two-probe measurements as part of a recent comparative study of diverse CB that included one freshwater CB enrichment culture (*Leonid et al., 2023*). Freshwater sediments can contain CB populations at densities comparable to their marine counterparts, and their activity strongly influences the sediment geochemistry (*Risgaard-Petersen et al., 2015*). Indeed, freshwater CB play an essential role in a cryptic sulfur cycle by stimulating sulfate reduction (*Sandfeld et al., 2020*) and have been harnessed to significantly reduce methane emissions from rice-planted soil (*Scholz et al., 2020*). From a materials perspective, investigations of freshwater CB may also uncover conductive biomaterials suited for applications in less saline environments that support higher electric fields than marine environments (*Sandfeld et al., 2020*; *Risgaard-Petersen et al., 2014*). Here, we report on the recovery, phylogenetic analysis, imaging, and electronic characterization of *Ca.* Electronema CB from Southern California freshwater sediments using atomic force and multi-probe transport techniques.

## Results
### Freshwater cable bacteria from Southern California sediments
#### Identification of cable bacteria sampling sites
We identified two freshwater sediment sampling sites for recovery and subsequent measurements of CB in the Los Angeles area. The first site, Ballona Freshwater Marsh (*Figure 1A*), is a shallow seasonal freshwater pond which is heavily vegetated by aquatic plants that supply sufficient organic matter to the pond bottom sediment in the wet season (December to early June), and is dried out from approximately late June to December every year. The sediment profile from the site was brown-orange-colored at the surface and dark in color with a sulfide smell below the surface, indicating a sharp oxic to sulfidic transition in the sediment. The second sampling site is Mungi Lake (*Figure 1A*), which is primarily fed by washes from surrounding areas in Eastern Los Angeles and therefore is precipitation

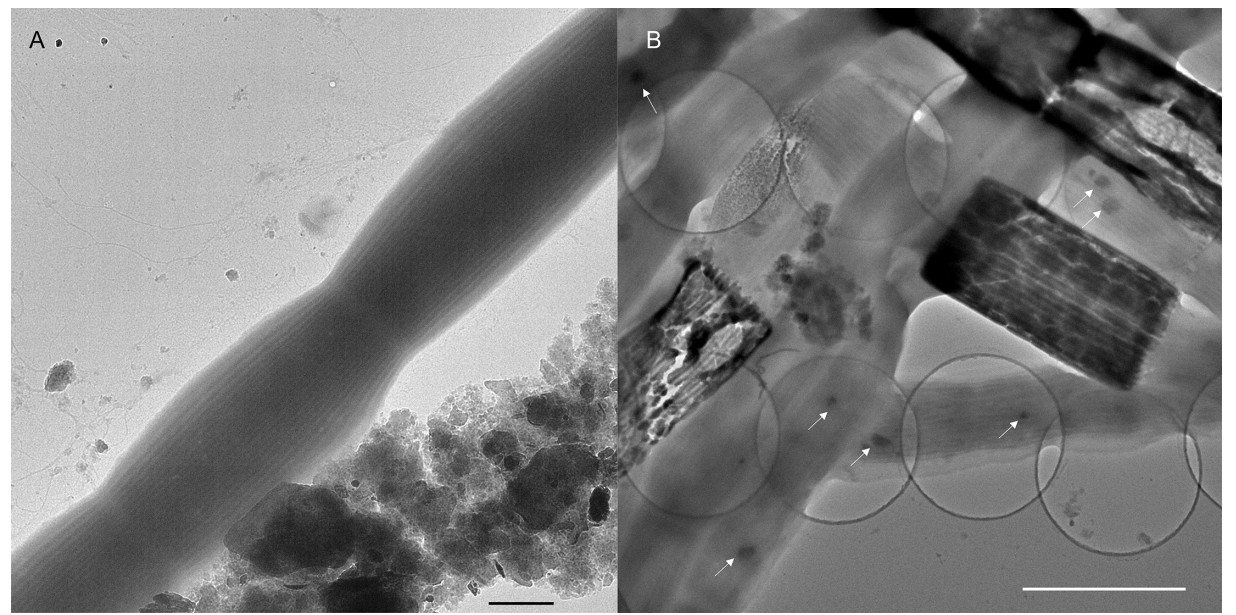

**Figure 2.** Transmission electron microscopy images of freshwater cable bacteria. (**A**) Transmission electron microscopy of Ballona Freshwater Marsh cable bacteria showing the characteristic cell surface ridge pattern. Scale bar: 1 µm. (**B**) The underlying network of parallel periplasmic nanofibers can be seen clearly in occasional degraded cells as the electron-dense scaffold of "empty cages". Small intracellular dark globules (white arrows) are consistent with polyphosphate granules commonly found in marine cable filaments. Scale bar: 2.5 µm.

dependent. Mungi Lake has a much thinner and more densely packed sediment layer, compared to the Ballona Freshwater sediment, as the lake was converted from a gravel pit, with almost no aquatic plants and therefore no roots in the bottom surface sediment.

## Southern California cable bacteria morphology

Thin silk-like filaments were directly observed by the eye within cracks of sediment at both sampling sites. Following laboratory incubation in circulating water to enrich for CB ('Materials and methods'), a high density of CB filaments was observed (*Figure 1B*). When observed with optical microscopy, the filament diameters varied from 0.7 to 3 µm, with thinner CB filaments usually observed within Mungi Lake sediments. Other than the diameter, there were no obvious morphological differences between the CB from the two sites. Occasionally, darker cells with slightly larger diameters than neighboring cells could be observed at seemingly random locations along the cable filaments (*Figure 1C*).

Transmission electron microscopy (TEM) imaging revealed that the freshwater CB display the characteristic previously observed parallel longitudinal periplasmic nanofibers that run along the entire CB filaments (*Figure 2A*). The estimated width of the fibers ranged from 49 to 57 nm, which is comparable to those of the marine cables (ca. 50 nm) (*Cornelissen et al., 2018*). TEM imaging of CB recovered from the sediments occasionally revealed degraded cells along the filaments where only an 'empty cage' of the parallel nanofibers remained, appearing as darker material reflecting a higher electron-dense composition (*Figure 2B*) and likely corresponding to the enlarged cells in *Figure 1*. These 'empty cages' were previously described as 'ghost filaments' which had lost all cell membrane and cytoplasm material (*Cornelissen et al., 2018*). Imaging of the freshwater CB also revealed small intracellular electron-dense globules, consistent with polyphosphate granules commonly observed in marine cables but not in freshwater CB previously (*Risgaard-Petersen et al., 2015*; *Kjeldsen et al., 2019*; *Sulu-Gambari et al., 2016*). The size of polyphosphate granules in freshwater CB is comparable or slightly smaller than in marine CB (*Sulu-Gambari et al., 2016*).

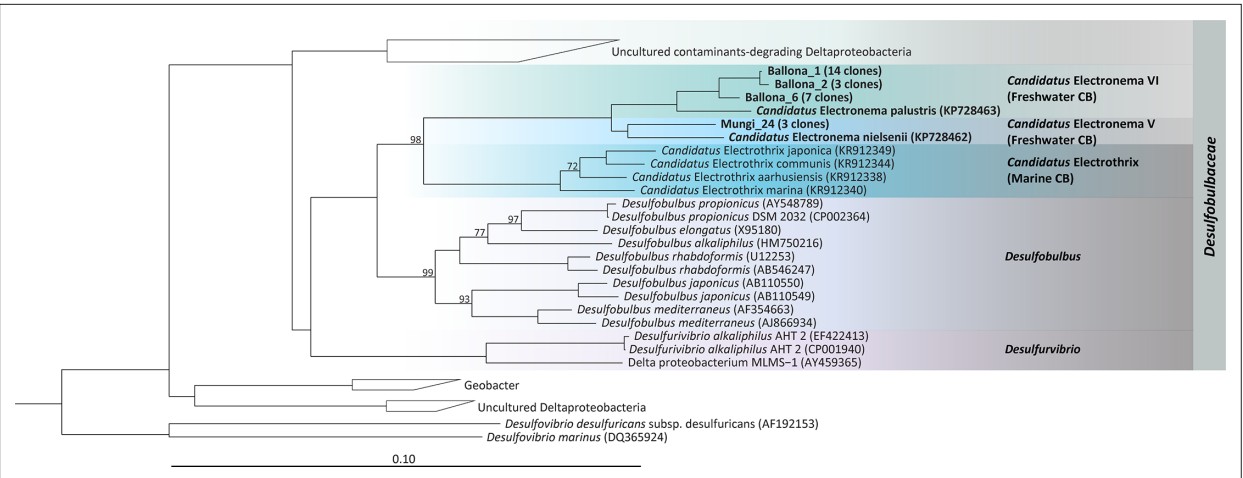

**Figure 3.** Phylogenetic tree based on 16S rRNA gene sequences of the Ballona Freshwater Marsh and Mungi Lake cable bacteria, other cable bacterial *Candidatus* taxa, and representatives of isolated members from the family *Desulfobulbaceae*. Scale bar represents 10% estimated sequence divergence. Bootstrap numbers (1000 re-samplings) > 70 are listed.

## 16S rRNA gene phylogeny of Southern California freshwater cable filaments and their associated bacteria

Three Ballona sediment CB filaments and three Mungi Lake sediment CB filaments were analyzed for 16S rRNA gene-based phylogenetic identification (*Figure 3*). All the Ballona sequences were within the family *Desulfobulbaceae*, and the majority (24/27 total Ballona sequences) grouped together with the freshwater CB sequence of *Ca*. Electronema palustris (96.7–97.3% similarity) (*Trojan et al., 2016*). The three representative sequences slightly differ from each other indicating coexistence of different species in Ballona sediment. The few non-CB sequences all affiliated to sulfate-reducing bacteria from freshwater sediments (*Sass et al., 1998*; *Ferrer et al., 2011*), with one sequence closely related to polycyclic aromatic hydrocarbon (PAH)-degrading bacteria from a novel genus *Desulfoprunum benzoelyticum* (*Junghare and Schink, 2015*).

The majority of Mungi Lake sequences affiliated within *Deltaproteobacteria* (26/35 clones). Three CB clones are closely related (>99% similarity) to uncultured sequences (accession number FQ658891 and FQ658831) from a PAH-contaminated wetland collecting highway road runoffs (*Martin et al., 2012*). These Mungi Lake CB sequences share only 96.5% similarity to the *Ca*. Electronema nielsenii and are even further distantly related to the sequences of Ballona CB and *Ca*. Electronema palustris. Most of the non-CB Mungi Lake clones (18/26 clones) grouped together to sequences identified from an altiplanic cold lake in South America (*Dorador et al., 2013*). Also in this group was a cluster (representative sequence: AJ389622, similarity 98.5%) resembling uncultured sulfate-reducers from a meromictic lake in Switzerland (*Tonolla et al., 2000*). Other Mungi Lake sequences affiliated to uncultured *Geobacter* clones (EF192881) from a temperate artificial lake that receives mining wastes, and a polychlorinated-dioxin-dechlorinating community that can also perform oxidative degradation of the dechlorinated products, enriched from a polluted river sediment enrichment (*Dorador et al., 2007*). Outside the *Deltaproteobacteria*, there were few sequences belonging to uncultured *Chlorobi* group (99.15% identity). One Mungi Lake clone was identified as *Ignavibacteria* (98.9% similarity to CP053446), which have been consistently found as the dominant flanking community in freshwater anammox bioreactors, presumably reducing nitrate to nitrite and degrading peptides, amino acids and EPS produced by anammox bacteria (*Ali et al., 2020*). One Mungi Lake clone shares 99.2% similarity to the uncultured subgroup 1 of *Acidobacteria* from an Australian pasture soil (2–4 cm) (*Sait et al., 2002*); other Mungi Lake sequences include one that loosely related to a potential *Hydrogendentes* bacterium (*Momper et al., 2018*) and four clones that reside deep in the branch.

# Electronic characterization of Southern California freshwater cable bacteria

## Electrostatic force microscopy

EFM was used to visualize the electric field variations on Ballona CB filaments using a dual-pass measurement approach. Topographic (*Figure 4A*) and phase (*Figure 4B*) tapping mode images of CB were generated during the first pass, revealing in high resolution the surface ridge pattern along two consecutive CB cells. During the second pass, performed with a tip voltage bias at a set height (50 nm) above the retraced CB topography, the phase image (*Figure 4C*) captured the electric force gradient distribution as a result of the change in the resonant frequency of the probe arising from the CB-tip electrostatic interaction. Consistent with a previous report (*Pfeffer et al., 2012*), the ridges and associated nanofibers exhibited conspicuously stronger electric forces relative to inter-ridge regions (*Figure 4C*), hinting at the unique electronic properties of the underlying nanofibers. The voltage dependence of the CB-tip interaction was also investigated, revealing that the phase difference between the elevated ridges and inter-ridge regions is proportional to the tip voltage squared (*Figure 4D*).

## Current–voltage measurements of freshwater cable bacteria across interdigitated array microelectrodes

For an initial assessment of freshwater CB conduction, current–voltage (I–V) measurements were performed along Ballona CB filaments bridging interdigitated array (IDA) microelectrodes in the oxygen-free environment of an anaerobic chamber (95% $N_2$, 5% $H_2$). In contrast to controls (open-circuit IDA and deposited drops lacking CB), which showed no detectable current, multiple devices with deposited CB showed linear up-to-μA currents along intact filaments in response to voltage in the –0.2 to 0.2 V range (*Figure 5*).

In contrast to previous observations of marine CB, which exhibited a rapid and massive decline in conductance upon exposure to air (*Meysman et al., 2019*), our observations of conductance in at least some freshwater CB filaments hinted at a higher degree of robustness upon oxygen exposure. For example, an IDA device with two CB filaments showed a modest slow increase in conductance during repeated I–V measurements over 3 hr in the oxygen-free atmosphere (*Figure 5*), ultimately reaching ± 3 μA in response to ±0.2 V. Upon removal from the anaerobic chamber and exposure to air, the conductance immediately declined to 80% of the previous value and slowly decayed until it ultimately stabilized at 63% of the oxygen-free value over 2.7 hr. When the IDA was moved back to the anaerobic chamber for further I–V measurements, the conductance again increased slowly until it stabilized after 2.4 hr (*Figure 5*). Interestingly, this behavior was filament-dependent; in another IDA that started with significantly smaller conductance under $N_2$ flow conditions (nA current values for 0.8 V), exposure to ambient air caused an immediate dramatic decline within minutes (*Figure 5— figure supplement 1*).

## Conductive atomic force microscopy: Point current–voltage spectra

The abovementioned conductance measurements probe transport longitudinally along multiple CB sections bridging IDA microelectrodes. We also performed point I–V conductance measurements through CB filaments between the underlying IDA and C-AFM tips; these measurements simultaneously probe longitudinal transport locally along specific CB sections and transversely across specific cells to the C-AFM tip above the filaments (*Figure 6*).

Tapping mode AFM images were first taken without voltage bias to visualize CB filaments laid on top of IDA microelectrodes and bridging 3–5 μm sections of the insulating quartz substrates (*Figure 6A–C*). No current response between the IDA and C-AFM tip was detected in response to applied voltage when the tip then probed the bare quartz surface near CBs. However, when the C-AFM tip probed points on the top surface of CB over quartz gaps, we detected currents up to tens of pA for voltages (–10 to 10 V) between the IDA and tip. These currents reflect a continuous path for electron transport from the underlying IDAs into cells, longitudinally along the conductive network of CB, and transversely into the overlying C-AFM tip. Interestingly, when such point I–V measurements were performed on CB exposed to air, the observed conductivity did not decay immediately, but slowly decreased over several hours, consistent with the IDA measurements reported above.

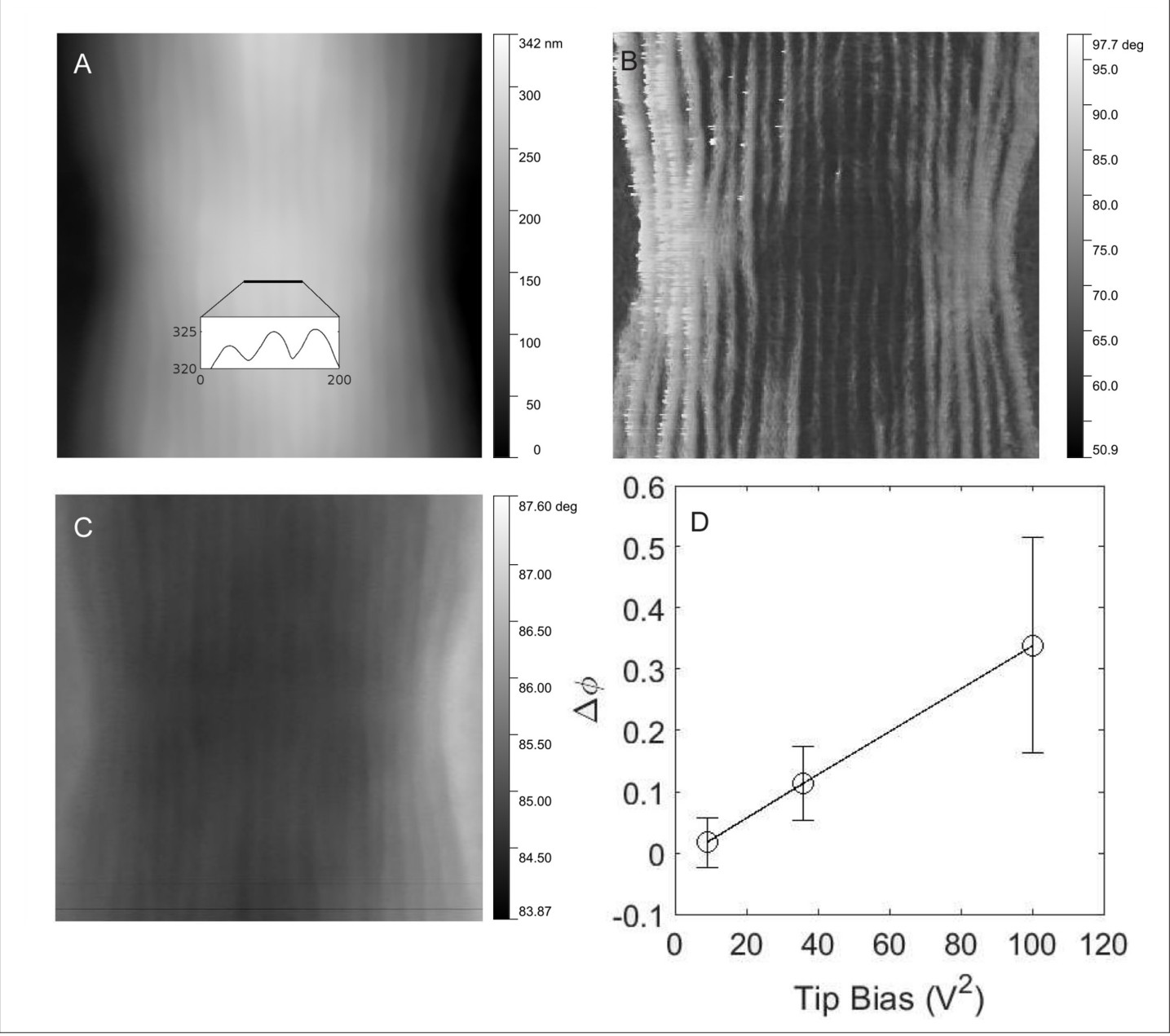

**Figure 4.** Electrostatic force microscopy on cable bacteria. (**A**) Tapping mode atomic force microscopy of freshwater cable bacteria near the junction between two cells. Inset is a cross-sectional line scan showing cell surface ridge pattern. (**B**) Atomic force phase image corresponding to the topographical scan and showing the ridge pattern in high contrast. (**C**) Electrostatic force microscopy image with a 6 V tip bias, collected from the retrace scan at 50 nm fixed height above the cable surface. The contrast in this image stems from the electric force rather than topography. (**D**) The voltage dependence of the electrostatic interaction (phase shift, Δφ, of the ridges with higher electric force relative to inter-ridge regions) between the conductive tip and the same cable bacterial filament from (**C**).

The online version of this article includes the following source data for figure 4:

**Source data 1.** Measured AFM height data with 10 V tip bias and 30 nm tip height.

**Source data 2.** Measured AFM napphase data with 10 V tip bias and 50 nm tip height.

**Source data 3.** Measured AFM phase data with 10 V tip bias and 50 nm tip height.

**Source data 4.** Measured AFM height data with 3 V tip bias and 50 nm tip height.

**Source data 5.** Measured AFM napphase data with 3 V tip bias and 50 nm tip height.

**Source data 6.** Measured AFM phase data with 3 V tip bias and 50 nm tip height.

*Figure 4 continued on next page*

*Figure 4 continued*

**Source data 7.** Measured AFM height data with 6 V tip bias and 50 nm tip height.

**Source data 8.** Measured AFM napphase data with 6 V tip bias and 50 nm tip height.

**Source data 9.** Measured AFM phase data with 6 V tip bias and 50 nm tip height.

## Conductive atomic force microscopy: Mapping the conductive network of cable bacteria

In addition to the point I–V measurements described above, C-AFM was used in contact scanning mode to image the electron transport path along CB hanging off microelectrodes under $N_2$ perfusion conditions (*Figure 7*). A constant voltage (5 V) was applied along the freshwater CB filaments between the microelectrode and an ultra-sharp (<5 nm tip radius) conductive single-crystal diamond C-AFM tip. Following a previous report on marine CB (*Thiruvallur Eachambadi et al., 2020*), a force set point was selected to gently scrape off the insulating outer membrane of the freshwater CB while allowing access to image the conductive network underneath. *Figure 7* displays a current map, stitched

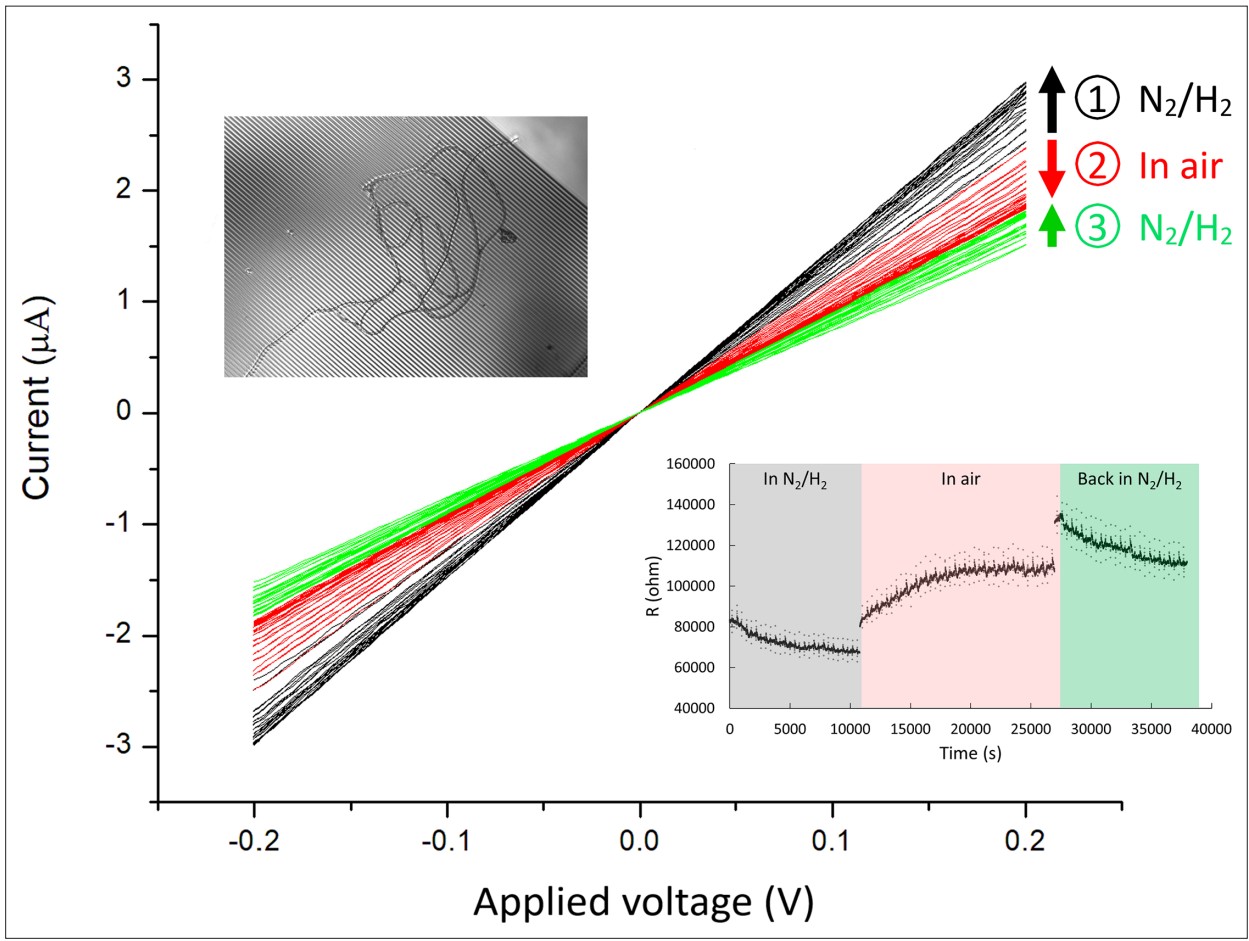

**Figure 5.** Current-voltage (I–V) measurements of Ballona Freshwater Marsh cable bacteria on gold interdigitated array (IDA) microelectrodes (3 μm gap between the fingers of the two electrode bands). For this measurement, two cable filaments were deposited on the IDA (inset). The I–V measurements (black) show a modest increase in conductance over time (3 hr) during consecutive measurements in an air-free environment (95% $N_2$ and 5% $H_2$). Upon air exposure (red), the conductance slowly declined over time (2.7 hr), and conductance recovered partially after re-insertion into an air-free environment (green). Arrow direction denotes measurements over time for a particular condition. The lower right inset demonstrates the resistivity change over time at different conditions.

The online version of this article includes the following source data and figure supplement(s) for figure 5:

**Source data 1.** An example of sensitivity of conduction in cable bacteria to air exposure.

**Figure supplement 1.** An example of sensitivity of conduction in cable bacteria to air exposure.

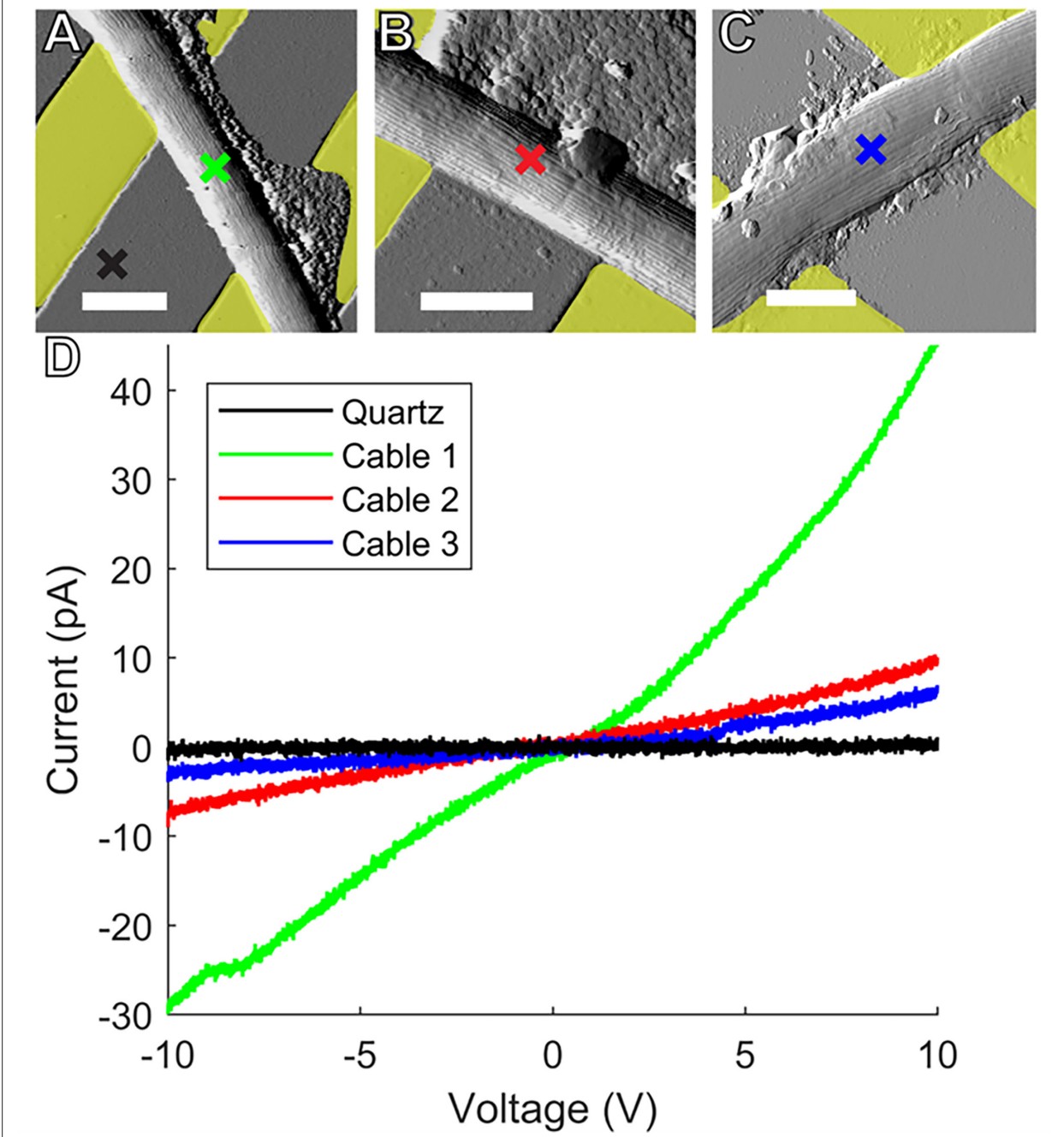

**Figure 6.** Conductive AFM point I-V measurements. Representative atomic force microscopy amplitude images of Ballona freshwater cable bacteria filaments spanning the insulating quartz gap between yellow false colored gold (**A, B**) and indium tin oxide (**C**) microelectrodes. Colored crosses mark the conductive tip location in each sample where point current–voltage (I–V) spectra were collected to measure electron transport between the microelectrodes and tip. Scale bars: 2.5 μm. (**D**) I–V responses from the three representative cables. Line color corresponds to the color of the corresponding cross mark in (**A–C**).

The online version of this article includes the following source data for figure 6:

**Source data 1.** The conductive point I-V data for the blue curve in *Figure 6*.

**Source data 2.** The conductive point I-V data for the black curve (on quartz) in *Figure 6*.

**Source data 3.** The conductive point I-V data for the green curve in *Figure 6*.

**Source data 4.** The conductive point I-V data for the red curve in *Figure 6*.

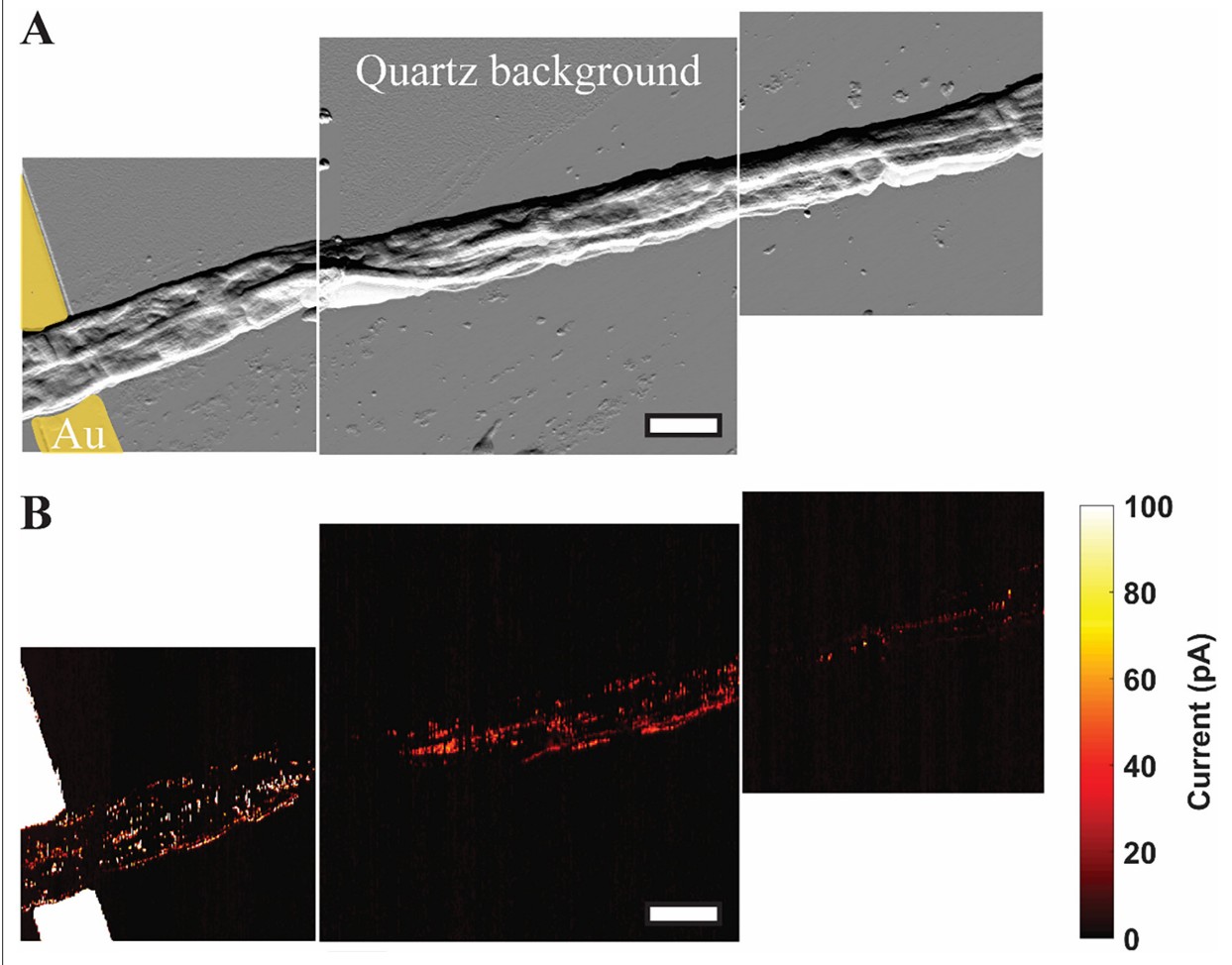

**Figure 7.** Mapping the conductive network of cable bacteria by conductive AFM. (**A**) A spatial montage of three consecutive tapping atomic force microscopy amplitude images of two parallel Mungi Lake freshwater cable bacteria in contact with, and extending away from, an electrode (false colored yellow). Scale bar: 3 µm. (**B**) Conductive atomic force current map images of the same sections shown in (**A**) with 5 V bias between the underlying electrode and conductive tip. The observed current signals largely correlate with the corresponding cell surface ridge pattern in (**A**).

The online version of this article includes the following source data for figure 7:

**Source data 1.** The amplitude data of the conductive AFM mapping.

**Source data 2.** The current data of the conductive AFM mapping.

**Source data 3.** The amplitude data of the conductive AFM mapping.

**Source data 4.** The current data of the conductive AFM mapping.

**Source data 5.** The amplitude data of the conductive AFM mapping.

**Source data 6.** The current data of the conductive AFM mapping.

together from three consecutive C-AFM images, of two CB filaments lying side-by-side and extending approximately 30 µm off a microelectrode. The imaging revealed parallel conductive paths consistent with the periplasmic nanofibers as the charge carriers in CB. Indeed, when compared with topographic imaging (*Figure 7*), the conductive paths coincided with the CB cell surface ridge pattern.

## Four-probe transport measurements

To quantify the intrinsic conductivity of freshwater CB and their periplasmic nanofibers, we measured electronic transport along Mungi Lake CB deposited on custom-fabricated four-probe (4P) microelectrode devices. In contrast to the two-probe IDA and C-AFM measurements described above, which include contributions from the CB-electrode contact resistances, the 4P measurements excluded

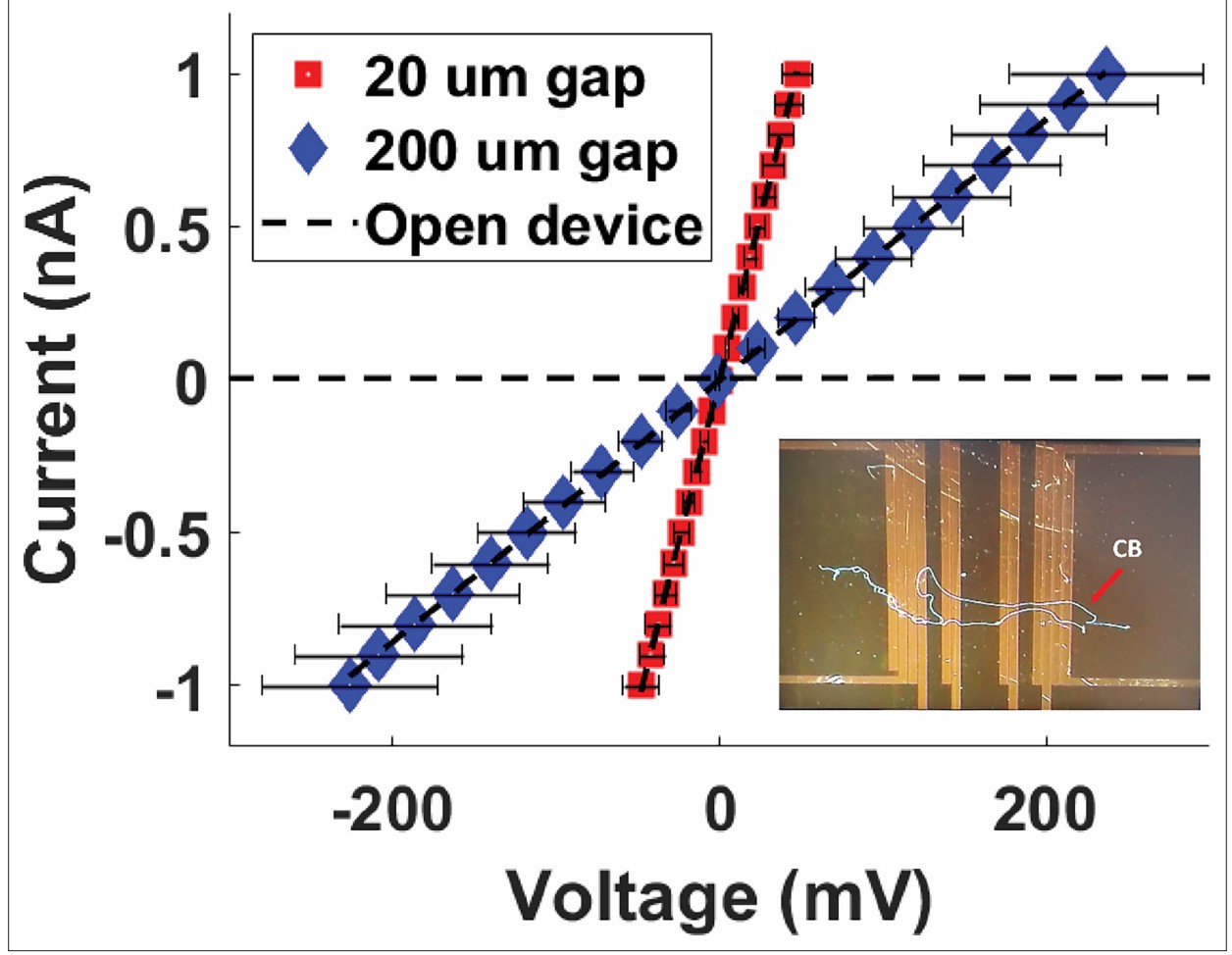

**Figure 8.** Four-probe current–voltage measurements of Mungi Lake freshwater cable bacteria filaments, shown for two different separations between the two inner probes. Inset: representative image of cable bacteria filaments across a four-probe device. Each data point is the average of three data points, and the error bars represent the standard deviation.

The online version of this article includes the following source data for figure 8:

**Source data 1.** Four-probe measurement 1 on the cable over a 200-micron gap.

**Source data 2.** Four-probe measurement 2 on the cable over a 200-micron gap.

**Source data 3.** Four-probe measurement 3 on the cable over a 200-micron gap.

**Source data 4.** Four-probe measurements on the cable over a 20-micron gap.

**Source data 5.** Four-probe measurement 2 on the cable over a 20-micron gap.

**Source data 6.** Four-probe measurement 3 on the cable over a 20-micron gap.

**Source data 7.** Four-probe measurement 1 on the cable over a 20-micron gap.

contact resistance by routing current between two external microelectrodes and separately measuring the voltage drop across two inner microelectrodes. Thus, our quantitative estimate of CB nanofiber conductivity was made only using 4P measurements, rather than C-AFM. The resulting 4P I–V plots (*Figure 8*) are shown for two separate inner probe separations, with the inverse slopes yielding whole CB resistances of 47 and 240 MΩ for the 20 and 200 µm lengths, respectively. By modeling the CB periplasmic nanofibers as current-carrying simple cylindrical wires forming a simplified circuit of parallel resistors, and taking into account each nanofiber's estimated diameter (50 nm) and number of fibers per CB (~35), we arrive at a single nanofiber conductivity on the order of 0.1 S/cm, from both gap sizes.

## Discussion

The appearance in many gene sequencing studies and increasing field observations suggest a cosmopolitan global distribution for CB in both marine and freshwater sediments (*Risgaard-Petersen et al., 2015*; *Burdorf et al., 2017*). Here, we identified two sampling sites (*Figure 1*) in the Los Angeles basin area, which served as a source for downstream phylogeny, microscopy, scanning probe, and electronic studies of freshwater CB. While CB have not been isolated into pure culture (*Thorup et al., 2021*), the freshwater CB from both Ballona wetlands sediment and Mungi Lake sediment could be reliably enriched in laboratory incubations to provide a continuous supply for studies (*Figure 1*). Based on 16S rRNA gene sequence phylogeny (*Figure 3*), CB from the heavily vegetated Ballona sediments are closely related to *Ca*. Electronema palustris (*Trojan et al., 2016*), which has been previously observed to be associated with the roots of aquatic plants (*Scholz et al., 2021*). On the other hand, CB from the non-vegetated Mungi Lake sediments are more closely related to those from PAH-contaminated sediment collecting water runoffs (*Martin et al., 2012*). The 16S rRNA gene analysis also revealed additional non-CB clones from both Ballona and Mungi Lake samples, many of which are highly similar to those from stratified and oligotrophic freshwater ecosystems (*Dorador et al., 2013*). Although the relationship and functions of these bacteria are unclear, they likely represent cells firmly attached to CB filaments, as all the filaments were thoroughly washed before DNA extraction. The communities from both sites have representative lineages related to organic contaminant degradation, which correlate to the environmental background of the two sites. In Mungi Lake, for example, various contaminants (chlordane, DDT, lead, dieldrin, PAH, and polychlorinated biphenyls) have been identified previously (*Environmental-Protection-Agency US, 2012*).

Morphologically, our freshwater CB filaments exhibited the expected characteristics of previously studied CB (*Pfeffer et al., 2012*; *Cornelissen et al., 2018*), including the cell surface ridge pattern and underlying periplasmic nanofiber network (*Figures 2–4*). The latter could be observed in TEM of freshwater CB both within intact cells and, occasionally, as the remaining cage scaffold from naturally degraded cells (*Figure 2*). The typical single nanofiber width encountered in our freshwater CB was 53 ± 8 nm, which is consistent with the previously reported nanofiber thickness from marine CB (*Cornelissen et al., 2018*) and recent characterization of a single-strain freshwater CB enrichment culture (33 nm) (*Leonid et al., 2023*). These results suggest that, while general morphological features are shared among diverse CB strains, the number and size of the ridges (and possibly their internal components) may depend on strain and environmental conditions. However, similarly to previous measurements of marine CB (*Pfeffer et al., 2012*), EFM captured stronger electric forces when scanning the cell surface above the ridges (*Figure 4*), again indicating the unique electric potential and charge distribution from the underlying conductive periplasmic nanofibers. By scanning the long-range electrostatic interaction between CB and tip at a fixed retrace height above the sample, we observed the expected voltage squared dependence of the EFM phase contrast (*Staii et al., 2004*), thereby unambiguously assigning EFM phase contrast to the electric force gradient rather than a topographic contribution. C-AFM with an ultra-sharp diamond tip that disrupts the insulating outer cell layer (*Figure 7*) offered an even more direct confirmation of the conductivity of underlying periplasmic nanofibers by observing conductive paths between the microelectrode and scanning tip that correlate with the CB ridge pattern. Furthermore, C-AFM point I–V spectra (*Figure 6*) confirmed a direct path for charge transport originating from the microelectrode below intact CB, along the internal periplasmic nanofiber, and into the C-AFM tip above the sample, consistent with the previously observed interconnectivity of the nanofiber network between cells (*Thiruvallur Eachambadi et al., 2020*). The latter measurements also illustrate a high degree of variability (*Figure 6*) in the conduction signal across different cables; it remains unclear whether this variability reflects heterogeneities intrinsic to different CB filaments or stems from culturing and/or sample preparation effects (including possible oxidative damage). Taken collectively, however, the EFM and C-AFM observations, performed here for the first time on freshwater CB, are consistent with previous analogous measurements on marine CB (*Pfeffer et al., 2012*; *Meysman et al., 2019*; *Thiruvallur Eachambadi et al., 2020*).

While their two-probe nature includes a contact resistance contribution, we found IDA microelectrodes to offer a convenient assessment of CB conduction properties (*Figure 5*) ahead of more sophisticated AFM and four-probe-based techniques (*Figures 6–8*). The IDA platforms, which are commercially available, have been previously used to investigate conductivity in bacterial biofilms (*Yates et al., 2016*; *Xu et al., 2018*) and result in high current signals (µA currents in *Figure 5*) as

a result of assessing transport along multiple CB sections spanning the IDA inter-electrode gaps. Interestingly, these measurements hint that freshwater CB conductivity may have a somewhat higher degree of tolerance to oxygen exposure than previously reported for marine CB. Instead of an instantaneous and dramatic decline in conduction upon exposure to air (**Meysman et al., 2019**), some of our freshwater CB samples exhibited a slow (hours) and only partial decline of the conduction upon exposure to air, which could be partially restored by re-insertion into an anaerobic chamber (**Figure 5**). While these observations showed a high degree of variability and therefore require a more detailed investigation, it is interesting to consider the possibility that the oxidative decline (or other damaging processes), thought to be a consequence of oxidation of Ni cofactors involved in electron transport (**Boschker et al., 2021**), may not affect all sections of the centimeter-long CB filaments simultaneously; under these conditions, IDA measurements, which probe multiple micrometer-scale electrode-crossing CB regions (e.g., 372 crossings in **Figure 5** inset), may offer an advantage over techniques addressing entire CBs or specific CB regions. It is also interesting to consider an alternative possibility that the conductive properties of freshwater CB may be intrinsically more oxygen-resistant than marine CB. A recent bioRxiv paper mentioned that although the cross-sectional area of a single fiber in a marine CB strain was ca. 50% bigger than a freshwater CB strain, the total surface of the fibers in the freshwater CB filament is 40% bigger than that of the marine CB filament. Meanwhile, the freshwater CB strain exhibit higher conductivity than the marine CB (**Leonid et al., 2023**). These differences hint at the possibility that the structure or morphology/size of the freshwater cables may be more robust against this oxidative degradation. In general, salt content affects the solubility of oxygen, which dissolves to 25% high levels in freshwater than saltwater. Indeed, previous CB studies verified 20–25% higher oxygen concentration at the surface of freshwater sediment (310–350 µM) (**Risgaard-Petersen et al., 2015**; **Sandfeld et al., 2020**), compared to marine sediments (200–280 µM) (**Pfeffer et al., 2012**; **Nielsen et al., 2010**; **Meysman et al., 2015**; **Schauer et al., 2014**; **Malkin et al., 2014**; **Sulu-Gambari et al., 2016**; **Risgaard-Petersen et al., 2012**; **Larsen et al., 2015**; **Seitaj et al., 2015**). Our results motivate a systematic investigation to correlate the oxygen availability in CB environments to both their conductivity and tolerance of electron transport to oxygen exposures.

When taking contact resistances of intact freshwater CB into account with four-probe measurements (**Figure 8**), we estimated the single nanofiber conductivity to be in the $10^{-1}$ S/cm range. Since CB may possess nonuniform conductivity along their length and estimates for nanofiber diameter and per cable amount were used in our single nanofiber conductivity calculation, we cannot constrain our reported value to better than on the order of 0.1 S/cm. Despite this, our figure falls within the $10^{-2}$–$10^1$ S/cm range recently reported for marine CB (**Meysman et al., 2019**; **Bonné et al., 2020**). We did not observe freshwater CB nanofiber conductivity near the upper end of the previously reported range (~80 S/cm). While these measurements are performed under ex situ conditions, they motivate the question of whether the measured conductivities are sufficient to support the in situ activity of CB in sediments. From previous measurements of the CB filaments density in sediment incubations (~$10^8$–$10^9$ m$^{-2}$) and the electron current densities linking electrogenic sulfur oxidation to oxygen reduction (74–96 mA m$^{-2}$) (**Meysman et al., 2015**; **Schauer et al., 2014**; **Geelhoed et al., 2020**; **van de Velde et al., 2017**), one arrives at a per cable current around 170 pA, which amounts to per nanofiber current of 5 pA given the average number of nanofibers seen in our cables. Assuming a driving voltage up to 1 V, representing the redox potential difference between the anodic sulfide oxidation and cathodic oxygen reduction half-reactions, for a typical nanofiber cross-sectional area and length (e.g., several millimeters), these conditions require a minimum nanofiber conductivity on the order of 0.1 S/cm. Remarkably, this heuristic calculation therefore sets a limit on conductivity that is consistent with the electronic measurements. It is also interesting to consider what the observed conductivities (**Meysman et al., 2019**; **Thiruvallur Eachambadi et al., 2020**; **Bonné et al., 2020**) and this study may teach us about the underlying physical electron transport mechanism in CB. It is common to interpret such extended transport in molecular systems in one of two limits: hopping between charge localizing sites such as redox factors (e.g., the hypothesized Ni/S groups of CB) or coherent band-like transport familiar from periodic systems including metals or semiconductors. In this context, the thermally activated nature of CB transport and the estimate of the room temperature electron mobility at ~ 0.1 cm$^2$/Vs (**Bonné et al., 2020**) are consistent with hopping transport, as the latter figure approaches but is still below the generally accepted minimum mobility (~1 cm$^2$/Vs) for band transport (**Polizzi et al., 2012**; **Pirbadian and El-Naggar, 2012**). Still, future electronic transport and structural studies (e.g., to

identify the precise molecular makeup of the nanofibers) are needed to resolve the underlying mechanism, particularly if conduction operates in a more complex regime than band theory or hopping, where electron transport may be delocalized transiently or permanently over several sites along the nanofibers (*Beratan, 2019*; *Beratan et al., 2015*).

## Conclusion

In summary, this study presents a detailed electronic characterization of *Ca.* Electronema CB from Southern California freshwater sediments. The cell envelope characteristics of the freshwater CB, including the characteristic periplasmic nanofiber network, are consistent with previous observations of the globally distributed CB. C-AFM and EFM confirmed the role of the nanofiber network as the conductive conduit of freshwater CB, as previously observed in marine CB. In addition, current–voltage measurements confirmed the conductivity of whole CB filaments while hinting that the conductivity of the freshwater filaments may be less sensitive to oxygen exposure than previously observed. The conductivity of the periplasmic nanofibers from four-probe electronic transport measurements was estimated at 0.1 S/cm, a value high enough to support the electron current densities linking sulfide oxidation to oxygen reduction through the electric metabolism of CB in sediment environments. While electronic characterization of this long-distance electron transport previously focused on marine cables, our measurements shed light on the conductivity of freshwater CB and highlight the potential applications of CB as bioelectronic conduits with both environmental and future biotechnological relevance.

## Materials and methods

### Sediment sampling and incubation

Surface sediment and pond water were collected from two sampling sites. The first site, the Ballona Freshwater Marsh pond (33°58′15″N, 118°25′51″W), is a restored wetland that was previously filled and farmed for decades and is located 18 miles from downtown Los Angeles. The pond is seasonal (accessible approximately December to early June during the wet season). The second sampling site is Mungi Lake (34°6′6.192″N, 118°0′44.2224″W), located in the Los Angeles River Basin in the city of Arcadia, CA, USA. The sediment samples were collected from shallow water because the dissolved oxygen (DO) typically declines to zero during the summer months at depths greater than 5 m (*Environmental-Protection-Agency US, 2012*). In the wintertime, less water stratification results in a higher DO concentration that can support CB growth in the lake bottom sediment. Mungi Lake samples were accessible year-round despite the lake shrinking during the dry season.

The top ~5 cm of sediment from both sites was sampled approximately every 3 months. Upon returning to the laboratory, the sediment was homogenized, sieved through a 5-mm-pore-size sieve, and carefully filled into glass jars, avoiding air bubbles in the sediment. The sediment jars were then incubated in a water bath at 15°C, submerged in circulating, air-saturated water collected from the respective sites.

### Cable bacteria filaments pickup

Filamentous bacteria were detectable after 2 weeks of incubation; occasionally, the filament density was high enough to be directly observed in sediment cracks (*Figure 1*). Over the incubation period, small sediment cores were taken from the sediment jars using plastic straws (8 mm diameter, 80 mm length) and transferred into small Petri dishes filled with DI water. Long filaments were hand-picked under a dissection microscope using glass hooks made from capillary tubes, then washed successively at least three times in Milli-Q water droplets. This extraction of CB was performed in a glove bag under an $N_2$ atmosphere.

### 16S rRNA gene sequence-based phylogenetic identification

Under a dissection microscope, three long filaments from each sampling site were washed and transferred into a sterile PCR tube. Cells were lysed by freeze-thaw three times as described in a previous study (*Schauer et al., 2014*). The lysates were directly used for PCR using the primer pair 8F and DSBB/1297R (*Kjeldsen et al., 2007*). PCR was performed according to the manufacturer's instructions for the Q5 Hot Start High-Fidelity 2X Master Mix (New England Biolabs, USA). After 30 s at

98°C for initial denaturation, PCR was performed with 35 cycles, consisting of 10 s at 98°C, 30 s at 65°C, and 45 s at 72°C, followed by a final step of 2 min at 72°C. The PCR amplicons were mixed and cloned using a Zero Blunt TOPO PCR Cloning Kit. Clones with a correct insertion were commercially sequenced (GENEWIZ, USA) using the vector primers M13F and M13R. Sequences were assembled in Sequencher 5.4, and their quality was checked manually. Alignment was performed in the ARB software package (arb-6.0.6) (*Ludwig et al., 2004*) using aligned sequences from SILVA database SSU 132 for reference (*Quast et al., 2013*) and manually refined subsequently. The 16S rRNA phylogenetic trees were generated using ARB. Trees that were built using the ARB neighbor-joining algorithm and the maximum likelihood algorithm (RaxML) with a 50% positional conservation filter, with 1000 bootstrap replicates in all trees, were compared to make sure the trees had no conflicting branching. Representative sequences from this study were deposited in GenBank with accession numbers (OR244071–OR244134).

## Transmission electron microscopy

Bundles of filaments were washed thoroughly, deposited in small droplets of Milli-Q water, and the CB filament-containing water droplets were then transferred and air-dried on PELCO hexagonal Carbon-Formvar-coated copper grids. The filament samples were unstained. High-vacuum TEM was performed using an FEI Talos F200C (Thermo Fisher, USA) Electron Microscope at 80 kV, and the images were captured using a CCD camera.

## Electrostatic force microscopy

EFM measurements were performed using an Oxford Instruments Asylum Research Cypher ES atomic force microscope (AFM). Substrates were affixed and electrically connected to sample discs. The CB were extracted from sediment using fine glass hooks, washed several times in DI water, and placed on the prepared substrates to dry. The sample discs were wired to the grounding pin of the AFM upon loading. Si probes with a Ti/Ir (5/20) coating, a nominal resonant frequency of 75 kHz (58–97 kHz range), a nominal spring constant of 2.8 N/m (1.4–5.8 N/m range), and a tip radius of 28 ± 10 nm were used for all EFM measurements. High-resolution EFM scans were performed following initial 'wide view' tapping AFM imaging to locate the CB filaments. For each EFM scan, a first pass scans the conductive probe in standard tapping without an applied tip bias, followed by a second 'nap' pass that retraces the first pass topography at a constant height above the sample. EFM was performed at different nap pass heights and tip voltage biases to arrive at optimal conditions and evaluate the voltage dependence. Electrostatic forces between the tip and sample shift the resonance frequency of the probe, resulting in a phase change of the cantilever, which was quantified for pixels representing specific features (e.g., nanofibers). The phase differences for each image (e.g., specific tip voltage bias) were averaged and their standard deviation taken.

## Current–voltage (I–V) measurements on IDA microelectrodes

A 3 µm gap-size Au IDA (IDA model# 012129, ALS Co, Ltd, Japan) and a Pine WaveDriver 200 Bipotentiostat/Galvanostat were used for electrical measurements. The IDA consists of two individually addressable arrays of interdigitated gold microelectrodes (length, 2 mm; width, 3 µm; thickness: 100 nm). Copper wires were affixed to these two electrodes with silver paint (product # 16035, TED PELLA, Inc) and covered with an epoxy layer at the contact points. The wires were then connected to the potentiostat, which swept the potential from –0.2 to 0.2 V at a 10 µV/s sweep rate. All measurements were performed in an anaerobic chamber (95% $N_2$ and 5% $H_2$) (Bactron 300, Sheldon Manufacturing, Inc, Cornelius, OR). Control measurements prior to measurements of CB included open circuit (without water or CB), I–V measurements of an IDA on which a water droplet has been deposited (no CB), and I–V measurements made after the Milli-Q water droplets were dried on the IDA (no CB). After these controls, CB filaments were washed and deposited in a small droplet of Milli-Q water on the IDA surface. The CB filament-containing water droplet was then left to dry prior to the I–V measurement.

## Conductive atomic force microscopy

C-AFM was performed using an Oxford Instruments Asylum Research Cypher ES AFM. CB filaments were placed and dried across the insulating gaps between Au (model# 012129, 3 µm gap) or ITO (model# 012128, 5 µm gap) IDA microelectrodes from ALS Co. Ltd. For C-AFM current mapping,

ultra-sharp conductive single-crystal diamond probes (AD-2.8-SS, Adama Innovations, Ltd), with <5 nm tip radius, were used to image Mungi Lake CB in contact mode under $N_2$ perfusion. C-AFM current mapping was performed with a 0.1 Hz scan rate, 300 nN force, and a constant 5 V DC voltage bias applied between the probe and IDA. The probes and imaging parameters were selected to minimize disturbance to the sample while simultaneously eroding the insulating outer membrane enough to electrically probe the underlying periplasmic fibers. Prior to C-AFM, high-resolution tapping mode imaging was performed to spatially correlate the topography of the CB nanofibers to conductivity.

Point current–voltage (I–V) measurements were also performed to measure conduction along Ballona Marsh CB between the IDA microelectrodes and C-AFM tip (Si Probe ASYELEC.01-R2, Oxford Instruments, with a conductive Ti/Ir [5/20] coating and 28 ± 10 nm tip radius) probing CB spots above the insulating quartz IDA gap under ambient conditions. At each point, the IDA voltage was swept for five consecutive cycles in the –10 to 10 V range with a 0.1 cycle/s scan rate, while the C-AFM probe was grounded and held in contact with the CB surface at a 450–510 nN force. All point–IV curves shown, including the insulating quartz background curve, are the average values taken from the five voltage cycles.

## Four-probe device fabrication

Four-probe (4P) microelectrode devices were designed in-house and fabricated by the University of California San Diego Nano3 cleanroom foundry service. Each device consisted of a series of parallel electrode bands. The bands were 3.5 mm long, 100 µm wide, and each one was connected on alternating sides to a 1.5 mm by 5 mm contact pad. The electrode band separation distances were variable, including multiple 20 and 200 µm gaps to assess different CB lengths. The fabrication process is briefly outlined as follows. 100-mm diameter and 0.5-mm-thick $SiO_2$/Si wafers (wafer model 1583, University Wafer, Inc) were used as substrates. Solvent-cleaned wafers were coated with photoresist (PR) and a laser writer was used to project repeats of the device pattern onto the PR with UV light. E-beam evaporation was used to deposit a 5 nm Ti adhesion layer followed by a 100 nm Au layer onto the wafers. Solvents were used to remove the excess Au/Ti layer and PR, leaving just the electrode patterns. As a final step, the wafers were coated in PR and each device was then diced into an 18 mm × 18 mm chip.

## Four-probe electrical measurements

Four-probe (4P) measurements were made using an Agilent 4156C Precision Semiconductor Parameter Analyzer, along with a Signatone 1160 series probe station. Thin copper wires (product # 1227, Train Control Systems) were electrically connected to the 4P contact pads with silver paint (product # 16035, TED PELLA, Inc) and alligator clips attached to tungsten probes were used to connect the 4P device to the analyzer. Mungi Lake CB filaments were placed and dried across the electrodes of our devices. To deal with the high degree of variability between individual filaments, they were placed one at a time and I–V measurements were performed after each deposition until current (from a conductive filament) was observed. To make the 4P measurement, a set of four electrodes on the device, with either a 20 or a 200 µm inner probe separation distance, were each wired to a source monitor unit within the analyzer. Current was swept between the outer two electrodes from –1 to 1 nA in step sizes of 100 pA. The voltage difference between the inner two electrodes ($V_{inner}$) was then recorded at each current step. Auto-range voltage sensitivity with a voltage compliance of 5 V and an integration time of 1 PLC were used. A 2 s delay was used between each $V_{inner}$ measurement to eliminate capacitive current contributions. Current sweeps were performed three times for the two inner probe gap sizes used. Average $V_{inner}$ values for each current step were obtained from the three sweeps, along with an error bar indicating the standard deviation. All 4P measurements were performed under an $N_2$ environment.

The single periplasmic fiber conductivity for the CB filaments measured at each inner electrode gap size was calculated from the formula $\sigma_{fiber} = \frac{L}{A\,N\,R_{cable}}$, where $L$ is the known spacing between the two inner electrodes, $R_{cable}$ is the CB filament resistance, $N$ is the average number of periplasmic fibers per CB filament, and $A$ is the average cross-sectional area of a single periplasmic fiber. An average N = 35 fibers per CB was determined from TEM imaging. $A$ was calculated from $\pi\left(\frac{d}{2}\right)^2$, with the periplasmic fibers assumed to be cylinders with average diameter $d$ = 50 nm estimated from TEM imaging. $R_{cable}$ was obtained as the inverse slope of best-fit lines for the first CB current vs. $V_{inner}$ scan. Since current

in each CB is carried by multiple periplasmic fibers, the $N R_{cable}$ factor, which represents the resistance of a single fiber, is based on a simplistic equivalent circuit model of a series of $N$ identical parallel fiber resistors. Open-circuit two-probe controls (for each pair of electrodes on the 4P device) were performed prior to placing CB on the devices.

## Acknowledgements

This work was supported by supported by the WM Keck Foundation award 8626 and Gordon and Betty Moore Foundation grant 10148. We thank Dr. Edith Read and the Friends of Ballona Wetlands organization for their kind help and sampling permission. We would like to thank the Nano3 cleanroom at the University of California, San Diego, for microfabrication services. TEM imaging was performed at the Core Center of Excellence in Nano Imaging at the University of Southern California.

## Additional information

### Funding

| Funder | Grant reference number | Author |
| --- | --- | --- |
| W.M. Keck Foundation | 8626 | Tingting Yang<br>Marko S Chavez<br>Christina M Niman<br>Shuai Xu<br>Mohamed Y El-Naggar |
| Gordon and Betty Moore Foundation | 10.37807/gbmf10148 | Tingting Yang<br>Marko S Chavez<br>Christina M Niman<br>Shuai Xu<br>Mohamed Y El-Naggar |

The funders had no role in study design, data collection and interpretation, or the decision to submit the work for publication.

### Author contributions

Tingting Yang, Marko S Chavez, Conceptualization, Data curation, Formal analysis, Investigation, Methodology, Writing - original draft, Writing - review and editing; Christina M Niman, Formal analysis, Investigation, Methodology, Writing - original draft; Shuai Xu, Investigation, Methodology; Mohamed Y El-Naggar, Conceptualization, Formal analysis, Supervision, Funding acquisition, Writing - original draft, Project administration, Writing - review and editing

### Author ORCIDs

Tingting Yang (iD) https://orcid.org/0000-0002-6644-841X
Mohamed Y El-Naggar (iD) https://orcid.org/0000-0001-5599-6309

Reviewer #2 (Public Review): https://doi.org/10.7554/eLife.91097.3.sa1
Author response https://doi.org/10.7554/eLife.91097.3.sa2

## Additional files

### Supplementary files
• MDAR checklist

### Data availability
Data generated or analysed during this study are included in the manuscript and supporting files.

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
