## [Editor Report · eLife assessment]

This work presents **fundamental** new insights into the conductivity of freshwater cable bacteria. The evidence supporting the conclusions, which was collected using appropriate techniques, is **compelling**. The work will be of interest to environmental microbiologists and the microbial electrochemistry community.

---

## [Referee Report · Reviewer #2 (Public Review)]

Summary:

In this work, Mohamed Y. El-Naggar and co-workers present a detailed electronic characterization of cable bacteria from Southern California freshwater sediments. The cable bacteria could be reliably enriched in laboratory incubations, and subsequent TEM characterization and 16S rRNA gene phylogeny demonstrated their belonging to the genus Candidatus Electronema. Atomic force microscopy and two-point probe resistance measurements were then used to map out the characteristics of the conductive nature, followed by microelectrode four-probe measurements to quantify the conductivity.

Interestingly, the authors observe that some freshwater cable bacteria filaments displayed a higher degree of robustness upon oxygen exposure than what was previously reported for marine cable bacteria. Finally, a single nanofiber conductivity on the order of 0.1 S/cm is calculated, which matches the expected electron current densities linking electrogenic sulphur oxidation to oxygen reduction in sediment and is consistent with hopping transport.

Strengths and weaknesses:

A comprehensive study is applied to characterise the conductive properties of the sampled freshwater cable bacteria. Electrostatic force microscopy and conductive atomic force microscopy provide direct evidence of the location of conductive structures. Four-probe microelectrode devices are used to quantify the filament resistance, which presents a significant advantage over commonly used two-probe measurements that include contributions from contact resistances. While the methodology is convincing, I find that some of the conclusions seem to be drawn on very limited sample sizes, which display widely different behaviour. In particular:

The authors observe that the conductivity of freshwater filaments may be less sensitive to oxygen exposure than previously observed for marine filaments. This is indeed the case for an interdigitated array microelectrode experiment (presented in Figure 5) and for a conductive atomic force microscopy experiment (described in line 391), but the opposite is observed in another experiment (Figure S1). It is therefore difficult to assess the validity of the conclusion until sufficient experimental replications are presented.

The calculation of a single nanofiber conductivity is based on experiment and calculation with significant uncertainty. E.g. for the number of nanofibres in a single filament that varies depending on the filament size (Frontiers in microbiology, 2018, 9: 3044.), and the measured CB resistance, which does not scale well with inner probe separation (Figure 5). A more rigorous consideration of these uncertainties is required.

Comments on revised version:

The authors address all of the comments carefully.

---

## [Author Response]

The following is the authors’ response to the original reviews.

**Reviewer #1 (Public Review):**
Summary:This work provides significant insight into freshwater cable bacteria (CB) and is an important contribution to the emerging CB literature. In this manuscript, Yang et al. describe currentvoltage measurements on CB collected from two freshwater sources in Southern California. The studies use electrostatic and conductive atomic force microscopies, as well as four-probe measurements. These measurements are consistent with back-of-the-envelope calculations on conductivities needed to sustain CB function. The data shows that freshwater CB have a similar structure and function to the more studied marine cable bacteria.Strengths:Excellent measurements on a new class of cable bacteria.Weaknesses:The paper would benefit from additional analysis of the data.
**Reviewer #1 (Recommendations for The Authors):**
This work provides significant insight into freshwater cable bacteria (CB) and is an important contribution to the emerging CB literature. In this manuscript, Yang et al. describe current-voltage measurements on CB collected from two freshwater sources in Southern California. The studies use electrostatic and conductive atomic force microscopies, as well as four-probe measurements. These measurements are consistent with back-of-the-envelope calculations on conductivities needed to sustain CB function. The data shows that freshwater CB have a similar structure and function to the more studied marine cable bacteria. Minor comments follow.

We are grateful to the reviewer for the encouraging feedback and for appreciating the central message of the preprint. Below we address the reviewer’s constructive comments.

Additional information could be provided regarding the degraded cells where an 'empty cage' remains, as well as the polyphosphate granules, which were previously observed in marine CB (refs. 11 and 18).

We have edited the manuscript to note that the appearance of empty cages and the polyphosphate granules in freshwater cable bacteria is indeed consistent with these features as previously reported in marine CB. The size of polyphosphate granules in freshwater CB are comparable or slightly smaller than in marine CB (Sulu-Gambari et al., 2015). In the case of empty cages, these cells were previously described as ‘ghost filaments’ which had lost all cell membrane and cytoplasmic material (Cornelissen et al., 2018).

Manuscript edits: a sentence regarding polyphosphate granules has been added into the manuscript from lines 307 - 308. “The size of polyphosphate granules in freshwater CB (70 nm – 400 nm) is comparable or slightly smaller than in marine CB (35)”.

A sentence regarding the empty cages has been added into the manuscript (lines 303-305). “These empty cages were previously described as ‘ghost filaments’ which had lost all cell membrane and cytoplasm material (20).”

The authors also state that the 'phase difference between the elevated ridges and interridge regions is proportional to the tip voltage squared,' and refer to Fig. 4D. This figure has only three data points with large error bars. The authors may wish to explain this finding and justify their analysis in greater detail.

We thank the reviewer for pointing out that we presented this result but did not adequately describe its origin or significance. In general, the probe phase response of electrostatic force microscopy (EFM) can originate not only from the electrostatic interaction with the sample (i.e. the electrical properties of interest) but also from shorter range van der Waals forces (which are more reflective of probe-sample distance i.e. topography). To ensure that EFM is reporting electrical interactions, we performed these measurements using a two-pass technique, with the second pass retracing the topography measured during the first pass, but at a fixed height above the surface where the interactions are long range (electrostatic) rather than short range (vdW) or resulting from topography cross-talk. The purpose of the voltage change measurement (Fig. 4D) is to simply assess whether this procedure is successful, since electrostatic forces are proportional to the square of the voltage at a fixed height (F = *½ . ∂C⁄∂z .V2*). While the error bar of that measurement is high, due to the intrinsic noise in the dynamic (high frequency) EFM phase response measurement, we note that the purpose of this measurement is simply to assess that the interaction is due to the electrical interaction with the sample, before proceeding to actual conductance measurements (Figs. 5-8).

Manuscript edits: we previously simply cited a reference where the reader can delve deeper into the origin of the square voltage signal. To put this into better context, we now include an additional information (lines 461 - 475), noting the origin and purpose of the result as described above.

It is interesting that the freshwater CB appear to be more resilient to air compared to marine CB (or at least some freshwater filaments, as the authors note that the level of resilience is filament-dependent). The authors indicate that salt affects oxygen solubility and there is a larger oxygen content in freshwater. Do the authors have thoughts on whether or not the differences between marine and freshwater CB could fit, or not fit, with the hypothesis that conductivity in air is lowered due to oxidation of the Ni/S species (ref. 25 in manuscript)? Could the freshwater CB have greater protection against oxidation?

We thank the reviewer for highlighting this point. Indeed, our manuscript mentions the current hypothesis that conductivity of cable bacteria may be diminished upon oxidation of the Ni/S groups (lines 101 - 105 and 498 - 504). It remains unclear how this idea may lead to variability between marine and freshwater cables. Interestingly, however, a recent comparative *bioRxiv* preprint (Digel *et. al.* 2023) noted significant differences in the morphology, number, and crosssectional area of nanofibers between a freshwater and marine CB strain. These differences may lead to a different resiliency against oxidative degradation upon exposure air. Specifically, even though the marine CB strain was characterized by a larger cross-section area per nanofiber, it had significantly fewer nanofibers, leading to 40% smaller total area than its freshwater counterpart. We have edited the manuscript to highlight these possible differences (at least in size) between freshwater and marine cables.

Manuscript edits (lines 506 – 514) “For example, a recent comparative study (21) hints at significant differences in the morphology, number, and size of nanofibers when comparing a marine CB strain to a freshwater CB strain. Specifically, while the marine CB was characterized by a 50% larger cross-sectional area per nanofiber, the total nanofibers’ area was 40% smaller than the freshwater strain due to a smaller number of nanofibers per CB filament. Given the proposed central role of nanofibers in mediating electron transport along CB, it is possible that such differences may also lead to different degrees of tolerance against oxidative degradation upon exposure to air.”

Figure 6D shows current-voltage measurements from three representative cables; there is a large variation, most notably between Cable 1 and Cables 2 and 3. Is this variation typical for different cables? Can the authors comment on the range of values observed and how many cables fit into different ranges? Any thoughts on the reasons behind the range?

Figure 6 B and C (red and blue) are representative of most of the cable conductance measured using the point IV CAFM technique, with the Figure 6 A (green) IV curve being an example of the upper limit, which was less frequently observed. In total we measured ten cables using the point IV CAFM technique. These variations may stem from actual differences in the conductivity of separate CB filaments, the environment of the measurement, or limitations in the conductive AFM measurement techniques. These limitations include a large contact resistance due to the interaction of the small probe with the sample, which may lead to large variability depending on the contact point. For this reason, we rely on 4-probe measurements (Fig. 8) for quantitative conductive analyses, rather than conductive AFM. It is important to note, however, that the conductive AFM measurements (Fig. 6 and Fig. 7) provide other complementary information including the demonstration of both transverse and longitudinal transport (lines 389-393) in Fig. 6 and the visualizing of the current carrying nanofibers in Fig. 7.

Manuscript edits: we have edited the manuscript (lines 413 - 418) to make it clear that the quantitative estimate of conductivity was made only using 4 probe measurements due to the limitations of CAFM or two-probe techniques.

Can the authors comment on how the number of fibers per CB in their samples compares with the number of fibers in marine CB? Marine CB are known to have pinwheel junctions where the fibers come together before branching out again. This pinwheel design could play a role in the function of the CB or in its survival (see Adv. Biosys. 2020, 4, 2000006). Were pinwheel structures observed in freshwater CB? If so, how do they compare?

From the previous studies, estimates of the number of fibers in marine CB appeared to vary significantly from 15 or 17 (Pfeffer et. al., 2012) to 58 – 61 (Cornelissen et. al., 2018). In our freshwater CB, we estimated the number of fibers at ~35 per CB (line 423), which is comparable to the count of 34 per freshwater CB recently reported by Digel *et al.,* bioRxiv 2023. We cannot specifically comment on the pinwheel structure as we did not perform the transverse thin section TEM imaging necessary to observe the cell-cell junctions in this particular study.

On lines 95-96, the authors discuss the fact that marine cable bacteria have a wide variance in their measured conductivities. While one may ask if the larger marine conductivities (near 80 S/cm) are representative, a conductivity of 0.1 S/cm is 2 orders of magnitude lower than this value, which the field generally refers to as a high conductivity. The authors should mention whether or not any of their specimens display the high conductivities seen in select marine cable bacteria specimens.

It is indeed important to note that the ~80 S/cm figure refers to an upper end previously observed (ref. 22) for marine CB conductivity. In our manuscript (lines 525 - 526), we highlight that the previously observed *range* (including in that same study) is 10−2-101 S/cm and we were careful to qualify the previously reported upper end with ‘reaching as high as’ (line 97). Note that this places our measurement of 0.1 S/cm within the previously reported range. We have not observed freshwater CB conductivity near the upper end of the previously reported range, and generally propose that these types of measurements are better analyzed in the context of the biological function rather than ‘high vs. low’. Towards that end, the manuscript (lines 527-537) makes the argument that the 10-1 S/cm figure may be sufficient to support the electrical currents mediated by CB in sediments. We have edited the manuscript to highlight that we did not observe single CB nanofiber conductivity near the upper limit previously observed in marine CB (lines 522 525).

**Reviewer #2 (Public Review):**
Summary:In this work, Mohamed Y. El-Naggar and co-workers present a detailed electronic characterization of cable bacteria from Southern California freshwater sediments. The cable bacteria could be reliably enriched in laboratory incubations, and subsequent TEM characterization and 16S rRNA gene phylogeny demonstrated their belonging to the genus Candidatus Electronema. Atomic force microscopy and two-point probe resistance measurements were then used to map out the characteristics of the conductive nature, followed by microelectrode four-probe measurements to quantify the conductivity.Interestingly, the authors observe that some freshwater cable bacteria filaments displayed a higher degree of robustness upon oxygen exposure than what was previously reported for marine cable bacteria. Finally, a single nanofiber conductivity on the order of 0.1 S/cm is calculated, which matches the expected electron current densities linking electrogenic sulphur oxidation to oxygen reduction in sediment. This is consistent with hopping transport.Strengths and weaknesses:A comprehensive study is applied to characterize the conductive properties of the sampled freshwater cable bacteria. Electrostatic force microscopy and conductive atomic force microscopy provide direct evidence of the location of conductive structures. Four-probe microelectrode devices are used to quantify the filament resistance, which presents a significant advantage over commonly used two-probe measurements that include contributions from contact resistances. While the methodology is convincing, I find that some of the conclusions seem to be drawn on very limited sample sizes, which display widely different behavior. In particular:The authors observe that the conductivity of freshwater filaments may be less sensitive to oxygen exposure than previously observed for marine filaments. This is indeed the case for an interdigitated array microelectrode experiment (presented in Figure 5) and for a conductive atomic force microscopy experiment (described in line 391), but the opposite is observed in another experiment (Figure S1). It is therefore difficult to assess the validity of the conclusion until sufficient experimental replications are presented.

We indeed acknowledge both in the abstract (line 23-26) and section 2.2 (lines 374-377) the variable nature of the sensitivity and filament-dependent response to air exposure. Our discussion (lines 498-506) considers the possible reasons for this variability:

‘While these observations showed a high degree of variability and therefore require a more detailed investigation, it is interesting to consider the possibility that the oxidative decline (or other damaging processes), thought to be a consequence of oxidation of Ni cofactors involved in electron transport (25), may not affect all sections of the cm long CB filaments simultaneously; under these conditions, IDA measurements, which probe multiple micrometer-scale electrode-crossing CB regions (e.g. 372 crossings in Figure 5 inset) may offer an advantage over techniques addressing entire CBs or specific CB regions. It is also interesting to consider an alternative possibility that the conductive properties of freshwater CB maybe intrinsically more oxygen-resistant than marine CB’.

To summarize , the manuscript points to the likelihood that the IDA technique used here may offer an advantage for detecting currents under damaging conditions since it interrogates multiple sections simultaneously. Furthermore, in a recent preprint from Digel *et al.,* (2023), the conductivity of the only freshwater strain investigated in that study was among the highest compared to other marine CB strains. Therefore, the freshwater CB being more resistant is one possibility to be investigated based on these observations and results. We therefore present the latter as a possibility in the discussion.

The calculation of a single nanofiber conductivity is based on experiment and calculation with significant uncertainty. E.g. for the number of nanofibers in a single filament that varies depending on the filament size (Frontiers in microbiology, 2018, 9: 3044.), and the measured CB resistance, which does not scale well with inner probe separation (Figure 5). A more rigorous consideration of these uncertainties is required.

The reviewer raises an important point. For these calculations, we made sure to determine the representative number of fibers per cable and thickness of the nanofibers (~50 nm) from our own samples. We indeed assessed the possible variability across our different cable filaments and found the fiber numbers varied from 30 – 44 (with 35 used as a representative figure in the paper). For the scaling of resistance with inner probe separation, our 4P results estimated that the CB resistances are 47 MΩ and 240 MΩ for the 20 µm and 200 µm lengths, respectively, rather than an expected tenfold difference if the cable has a uniform conductivity along the entire filaments. This result suggests nonuniform conductivity in different sections of the CB filament. Since accounting for non-uniform conduction (and variability in fiber morphology/density) is clearly difficult, we were careful to limit our conclusion to an order of magnitude estimate (e.g. lines 522-525). Given the previously reported range of cable bacteria conductivity (10−2101 S/cm), this places our estimate within this range. We have further edited the manuscript to note that our reported single nanofiber conductivity cannot be constrained further than the order of 0.1 S/cm due to our estimates in nanofiber diameter and per cable amount as well as the possibility of nonuniform conductivity along the CB length (lines 522-525).

**Reviewer #2 (Recommendations for The Authors):**
Figure 4A: Please add scale- and color bar.

Done - new Fig. 4 included with colors bars for topography and phase. The inset of Fig. 4A denotes a 200 nm scale bar (and that scale is now mentioned in the figure caption)

Figure 5: A time series graph might be more instructive.

Done - we indeed appreciate this suggestion and find that it improved the clarity of Figure 5. An inset has been included in Figure 5 plotting the resistance R change over time under different conditions. This inset demonstrates that the resistance of the cable on the IDA was slowly decreasing in the N2/H2 anaerobic chamber, only to start increasing upon exposure to ambient air.

After putting the cable back into the chamber, the resistance again decreased over time.